# Milled Surface Integrity: Application to Fixed Dental Prosthesis

**Nicolas Lebon [1,2,]\* and Laurent Tapie [1,2]**

1   Unité de Recherche en Biomatériaux Innovant et Interfaces-UR 4462, Université Sorbonne Paris Nord, 93000 Bobigny, France; laurent.tapie@u-paris.fr
2   Prosthodontic & Biomaterial Departments, Université de Paris, 92120 Montrouge, France
\*   Correspondence: lebon@sorbonne-paris-nord.fr; Tel.: +33-(1)-58-07-67-85

**Abstract:** Surface integrity is a multiphysics (biological, mechanical, optical, chemical, esthetic, etc.) and multiscale (from nm to mm) concept. It is defined as the residual signature left on the surface by the manufacturing or post-treatment process and permits correlating the process with the expected surface functionalities. Thanks to the advances made in mechanical engineering, the concept of surface integrity has been transposed to dentistry and oral science. The surface integrity concept transposed to fixed dental prostheses is presented in this article. The main components of surface integrity and their correlations within the triptych of surface integrity–process–clinical functionalities are presented.

**Keywords:** surface integrity; dental prosthesis; manufacturing; functionalities



## 1. Introduction

In the framework of dental treatments, the restoration of dental functions using prostheses is a major challenge. The evolution of manufacturing techniques used in dental restorations involves the direct transposition of technological advances made in industrial engineering. Thus, in 1973, the foundations of computer-aided design and computer-aided manufacturing (CAD-CAM) applied to dentistry were laid by F. Duret. The purpose of using CAD-CAM is to replace so-called conventional molding techniques by a fully digitized prosthesis manufacturing process. The design of the prosthesis and its physical production by machining became wholly assisted by a computer. This modern technique of producing fixed prostheses, using industrial methods, makes use of raw blocks of biomaterials. These ceramic blocks, made of polycrystalline ceramics, glass ceramics or composite hybrid materials (polymer-infiltrated ceramic network (PICN)), are shaped by removing material using computer numerically controlled machine tools (CNCMTs). The quality of prostheses expected by patients and practitioners is therefore a major challenge for public health, since today 50% of the European population possess a dental prosthesis. The most usual aspects of quality are: esthetics, mechanical durability, non-adherence of bacterial plaque, biocompatibility and micromechanical retention of the prosthesis on the remaining dental tissues. These quality characteristics, satisfying prosthetic functionalities, require different physical and geometric components on the prosthetic surface, linked to the material and its shaping. In engineering, all these components can be grouped under the concept known as surface integrity. Following the presentation of the surface integrity concept, this article is devoted to its transposition to dental prostheses. By relying on the experience acquired in mechanical engineering, this transposition then allows for establishing correlations between surface integrity, expected prosthetic functionalities by practitioners and patients, and the machining process implemented in dental CAD-CAM manufacturing procedures.

## 2. Surface Integrity Concept

### 2.1. Conceptual Definition of Surface

From the viewpoint of materials engineering, a surface is an area separating an object from its surrounding environment. It corresponds to an interface (solid–gas, solid–liquid, solid–solid) having physical and chemical properties that depend on the composition of the material and the environment. The chemical composition and mechanical properties of this area can differ considerably from those of the bulk of the material. The term "surface" represents a layer with a thickness ranging from $10^{-2}$ to $10^{-6}$ mm [1,2].

A surface in the context of mechanical engineering is defined as the entire external surface and the sublayers forming a gradient of functionalities providing the best possible cost/benefit ratio [3]. Since the surface is not limited to the external layer [1], the near sublayers also play an important role in relation to some functionalities expected from a manufactured part.

### 2.2. Conceptual Definition of Surface Integrity

In the fields of material and mechanical sciences and engineering, the concept of surface integrity is often used to describe a surface that has just been manufactured. The term "surface integrity" therefore represents an understandable characterization of all the factors influencing the properties of the surface and the functionalities of the finished part's shape [4]. Thus, surface integrity is a highly multidisciplinary notion [3].

Two definitions of surface integrity are identified in the literature. First, the term surface integrity is defined as the residual signature left by the manufacturing process on the surface produced, influencing the quality of the surface obtained [5]. The second definition specifies that the integrity of the surface can be characterized according to different components at the surface and in the sublayers directly linked to the functionalities expected (optical, biological, chemical, electric, mechanical, thermal, etc.) [6].

In conclusion, the surface integrity concept relies on the residual signature left by the manufacturing process. It is represented by several components (geometric and physical) characterizing the surface and its sublayers. These components characterize the surface functionalities (optical, mechanical, chemical, esthetic, thermal, biological, electric, etc.). Thus, surface integrity is a concept allowing the establishment of correlations between surface manufacturing processes and surface functionalities.

## 3. Surface Integrity Specification

As mentioned above, the functionalities linked to surface integrity are represented by several components.

– A component is a group of several different parameters of the same nature and scale of observation.
– A parameter is an elementary constituent of a component that allows for representing a property (physical, mechanical, optical, thermal, biological, electrical, etc.). It can be qualitative or quantitative. A parameter is quantified by one or more indicators.
– An indicator is the numerical value associated with the corresponding parameter. Therefore, there is no numerical indicator for a qualitative parameter.

The structural organization of surface integrity is pyramidal (Figure 1). The components are grouped in the first level. The second level comprises the parameters linked to each component. The last level comprises the indicators corresponding to each parameter, if any.

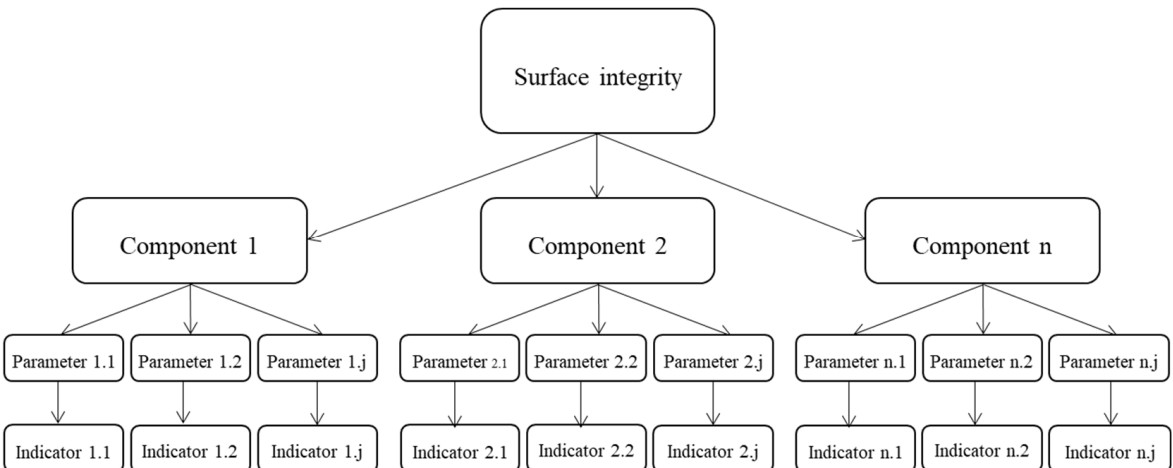

**Figure 1.** Structural organizational of surface integrity.

The functionalities linked to surface integrity are found both on the surface and in the core of the shaped part. It is therefore possible to define two categories of components: surface components (external surface integrity) grouping the surface integrity components involved in the surface (as understood by the material science definition of a surface), and the components of the sublayers close to the shaped surface (internal surface integrity) grouping the components involved in sublayers of the surface (as understood by the mechanical engineering definition of a surface).

### 3.1. External Component Classification

The two identified components of surface integrity linked to the external surface are only geometric: roughness and edge quality.

#### 3.1.1. Roughness

The roughness component, expressed in μm, characterizes the microgeometry of surface topography. There are several standards relating to 2D roughness [7–9]. More recently, the importance of 3D roughness parameters in scientific applications, since surfaces are 3D by nature, has been established. Therefore, the use of 3D roughness parameters results in a pertinent concept compared to 2D profilometry. In addition, the characterization of 3D surfaces is useful when the topography is anisotropic or presents singularities (holes, peaks, pits) [10]. By extension of 2D parameter standards, standards corresponding to 3D parameters are available [11,12]. They quantify a surface and not only a 2D profile. Standard EN 623-4 [13] specifically sets out the techniques to be used and the precautions (linked to optical reflection problems) to be taken at the moment of measuring, without contact, the roughness of technical ceramics similar to those used in restorative dentistry.

#### 3.1.2. Edge Quality

Edge quality comprises two parameters linked to machining localized to sharp edges such as the dental prosthesis cervical margin. The presence of burrs is the first parameter of the machined edge. Material removal operations always lead to the formation of burrs, at a micro- or macroscopic scale, located on the edges of the part. They are due to plastic deformation occurring during the machining process. This first parameter of edge quality is a qualitative parameter based on the more or less pronounced presence of burrs [14].

The second parameter of edge quality is chipping. In dentistry, as well as for other medical or industrial applications, ceramic materials like quadratic zirconia stabilized by yttrium oxide (Y-TZP), are well known to be sensitive to chipping when shaped by a mechanical machining process. Chipping is characterized by a significant loss in the volume of material (chip). In the dental prosthesis machining context, a parameter called "chipping factor" (CF%) has been defined to characterize the rate of chipping at the cervical

edge [15,16]. The associated indicator is the ratio of the length of the edge presenting chipping along the total length of the edge. The CF one-dimensional parameter does not take into account the width and the depth of the chip, only its length. It can be a drawback as the three dimensions of chipping can affect the expected clinical functionality in a marginal limit area.

### 3.2. Internal Components

The sublayers are affected by a fairly large number of components. They are microstructural (microstructural modifications, open porosity) and mechanical (internal stresses, microhardness, damage to the sublayers).

### 3.2.1. Internal Residual Stress

The shaping of the raw material, heat treatments and machining generate a complex combination of thermal, mechanical and chemical stresses at the surface and in the sublayers of the part [17–19]. The part is partially freed from these stresses after machining. Since they cannot be totally eliminated, the remaining stresses are dissipated in the form of plastic deformation, leading, in particular, to stresses at the surface and in the sublayers. These residual stresses in the material are called internal stresses. Internal residual stress can therefore be defined as the stress (traction or compression) present in a piece free of any external stresses. Mechanical loads (pressure and shearing), similar to those present in machining, generally lead to residual compression stresses due to plastic deformation in the material at the surface and in the core [20,21]. Generally expressed in MPa, the internal stresses present in a polycrystalline material depend on the history of stresses (mechanical, thermal, etc.) and they self-balance. The residual stresses can be classed into three orders (types), related to the observation magnification [22]. Residual stresses of the first order (type I), also called macrostresses, are homogeneous over a very large number of crystal domains of the material. The internal forces related to this stress are balanced on all planes. The moments related to these forces are equal to zero around all axes. Residual stresses of the second order (type II), are homogeneous within small crystal domains of the material (a single grain or phase). The internal forces related to these stresses are in balance between the different grains or phases. Residual stresses of the third order (type III) are homogeneous in the smallest crystal domains of the material (over a few interatomic distances). The internal forces coupled to these stresses are in balance in very small domains (such as around dislocations or point defects). Type II and III residual stresses are collectively termed microstresses. In the case of real materials, the actual residual stress state at a point comes from the superposition of the three stresses.

### 3.2.2. Microhardness

The hardness of a surface characterizes the capacity of a material to resist penetration. There are several hardness parameters (Vickers, Brinell, Knoop, Rockwell, etc.) defined by standards and depend on the materials tested [23–25]. Each parameter has a different hardness scale, though they are all based on measuring the dimensions of the impression left by a hard penetrating object (diamond) under load in the material. Microhardness tests, for which much smaller penetrators and lower loads are used, are usually employed to evaluate the local hardness of different sublayers. There are two microhardness tests: Vickers and Knoop. The hardness/microhardness indicator is therefore equivalent to a pressure.

### 3.2.3. Open Porosity

Porosity is a property of a material that presents interstices between its particles or grains (porous material). Open porosity means that the interstices form a network of pores that communicate with each other. The open porosity indicator is the ratio in percentage of the volume of voids (in and between the grains) over the total volume (apparent volume) [26]. As a function of the application, porosity can be desired or not

and obtained by the shaping process (for example, sintering of ceramics, powder-based additive manufacturing) of the material.

### 3.2.4. Microstructural Modifications

At different scales, several surface sublayer integrity parameters are linked to the microstructure of the material.

Phase transformations exist at the grain scale of a material. These are changes in the crystalline structure caused by temperature and/or high pressures occurring during machining. Phase changes are quantified by a rate of transformed volume.

Dislocations are present at the scale of the crystalline structure. These are linear defects corresponding to a discontinuity in the organization of this structure. At the nanometric scale, dislocations are characterized by a surface density indicator ($m^{-2}$).

### 3.2.5. Damage to Sublayers

The parameter linked to sublayer damage is cracking. It is characterized by discontinuity defects in the material. The particularity of a crack is that it has a very sharp point. Cracks, whether emerging or not, appear between the joints of grains. The length of a crack is noted "a" in microns. Cracks are defined as having a depth/width ratio higher than or equal to 4. Macrocracks are defined as being observable by a maximum magnification of 10X, whereas microcracks require more powerful magnification [27]. Damage to the sublayers reduces the mechanical characteristics of the part.

### 3.3. Component/Parameter Interactions

Whether the components are external or internal to surface integrity, they are not totally independent and interactions occur between them. These interactions are also present at both the microscopic and macroscopic scales. Apart from polymers, dislocations occur that generate internal stresses (in traction or in compression) [28]. Several models characterizing the dislocation–internal stress relation have been developed [29,30].

The internal stresses generated can result in phase transformations within the material. Generally, these transformations do not occur at a constant volume, and new internal stresses may occur [31,32]. Above a certain threshold, internal stresses become too high for the material, leading to cracking [33]. The location and size of the cracks generated by internal stresses depend on their distribution within the part. Thus, the cracks generated again cause internal stress concentrations at the tip of acute cracks. In the case of zirconia, this phenomenon is rather specific and is known as a reinforcement mechanism.

Roughness is also a potential source of cracks. Indeed, when the bottoms of valleys in the roughness topography reach a critical size for the material, cracks may occur and propagate [33–35]. The critical size of a crack $a_c$ for a given material is predictable [36]. When the cracks reach the surface of the piece, the result is an increase in open porosity, often leading to chipping of the surface [37].

Finally, the characterization of surface integrity relies on a number of components and multiscale and multiphysics parameters that interact with each other. The seven components of surface integrity (two relating to the surface and five to the sublayers) are observed at scales ranging from the millimetric to the nanometric [38]. Each component has at least one parameter. Figure 2 summarizes the hierarchy of surface integrity, components, parameters and indicators.

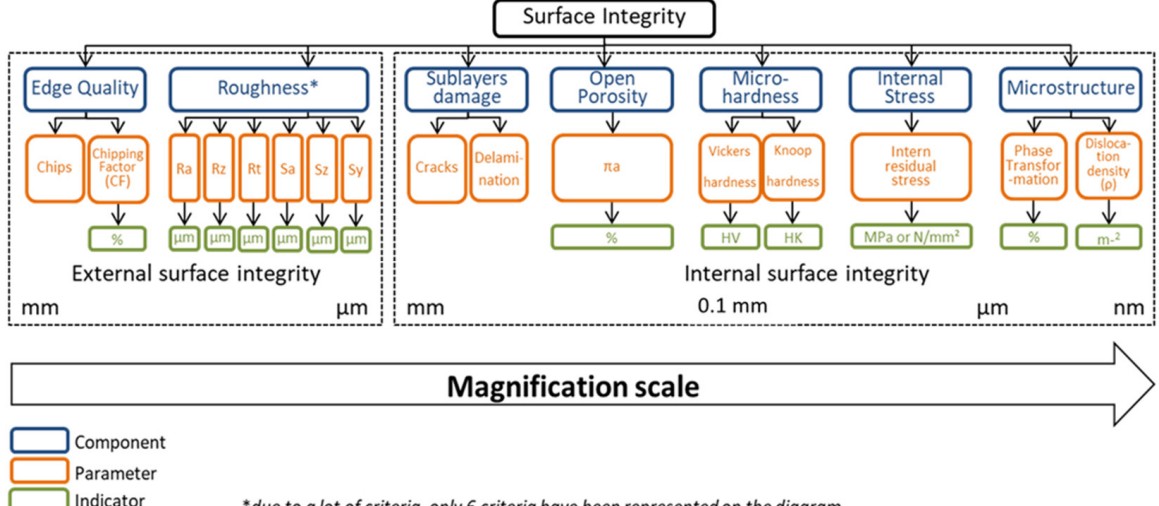

**Figure 2.** Structural diagram of dental surface integrity.

## 4. Process–Surface Integrity–Functionalities Triptych

It is impossible to establish direct relations between the manufacturing process and the functionalities expected on the surface (Figure 3, Correlation II). Initially, surface integrity is correlated with the parameters of the manufacturing process (Figure 3, Correlation Ia), leaving a characteristic signature on the surface. Secondly, surface integrity is correlated with the expected functionalities (Figure 3, Correlation Ib). This correlation permits the translation of the functionalities expected on a surface into indicators associated with the parameter of surface integrity.

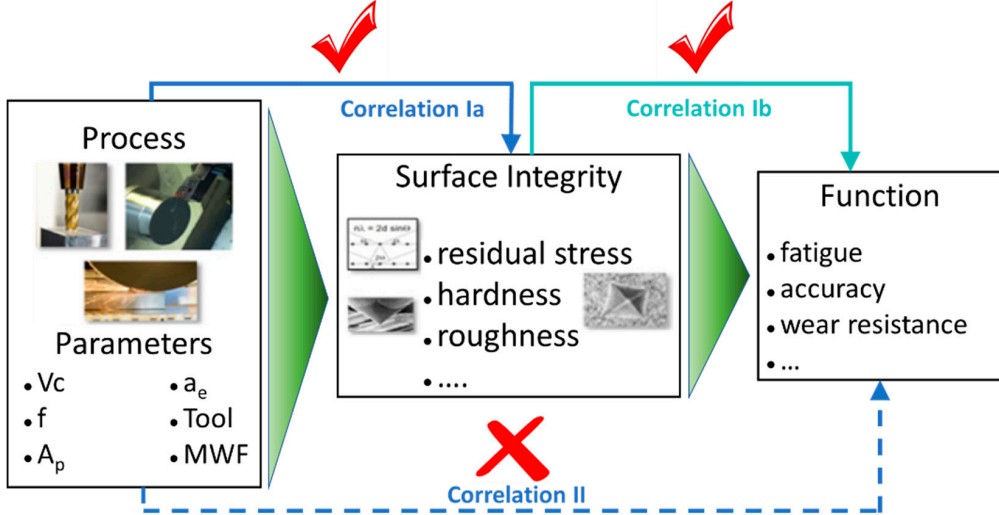

**Figure 3.** Possible correlations between the process, surface integrity and the expected functionalities [17].

Controlling the surface integrity of fixed dental prostheses requires further investigation into these two correlations.

### 4.1. Process–Surface Integrity Correlation

As defined above, surface integrity can be considered as a residual signature left by a manufacturing process on the surface produced and influencing the quality of the surface obtained. The analysis of this residual signature can be a pertinent tool for understanding the mechanisms responsible for generating surface and subsurface properties [39].

Many scientific publications have focused on machining in the field of mechanical engineering. In view of achieving its transposition to the field of dentistry, the bibliographic study of the "process–surface integrity" correlation is limited to a few materials and to the two processes most similar to those used in restorative dentistry. Regarding the materials selected, we focus on the metallic materials (steels, titanium alloys, aluminum alloys) and ceramic materials (alumina, zirconia) close to prosthetic restoration materials. The first process selected is milling by cutting with a ball-end mill cutter. This process and tool geometry are similar to those used in dental CAD-CAM. The second, abrasive grinding, is also quite similar to dental CAD-CAM. In a review of the literature, Benardos et al. [40] identified the factors affecting roughness during the machining of metallic materials. They found four families of factors affecting roughness: the intrinsic characteristics of the raw part (its hardness and its size), the characteristics of the cutting tool (geometry and material), the machining parameters (feed rate and cutting speed, tool tilt angle) and the cutting phenomenon (strength, vibration, chip formation).

Several authors agree that roughness and the internal stresses obtained when milling metallic materials with a ball-end mill cutter are closely linked to the tilt angle of the tool in relation to the machined surface [41–44]. The type of machining used is known as top milling (the tool axis is collinear to the machined surface normal) and flank milling (the axis of the tool is perpendicular to the machined surface normal). In top milling, the tool end works where the effective cutting diameter is almost null, or null. In flank milling, the tool works in the lateral zone, where the effective cutting diameter is close to the nominal diameter. An isotropic surface topography associated with considerable roughness and residual internal compression stresses are obtained with top milling. These results can be explained by the null cutting speed on the tool axis, generating friction and the phenomenon of ploughing in the cutting zone. Lower roughness and internal compression stress as well as an anisotropic surface topography are observed for higher tool axis angles in relation to the surface normal, providing higher cutting speeds and less friction.

The microstructure of metallic materials is deformed by top milling, contrary to other angles. These deformations are caused by the pressure exerted by the cutting tool, which are higher with top milling [41–45]. No significant differences in the surface hardness of metallic materials milled with a ball-end mill cutter were noted, whatever the angle of the tool in relation to the surface normal [42,43].

Whether milling is done by cutting or grinding, an increase in the feed rate generates an increase in roughness and reduces internal compression stresses. The relation between roughness and feed rate does not appear to be linear [43,44,46,47]. Cutting speed also has a significant impact on roughness and chip formation. Whether milling metallic materials by cutting or by grinding, an increase in cutting speed reduces roughness, but leads to more burrs on the sharp edges of the part [14,46,48,49].

Grinding different materials shows that the size of the diamond grains of the tool and the machined material impact the roughness and density of dislocations (microstructure) [47,49,50]. The smaller the diamond grains are, the lower the roughness is. Damage to sublayers by grinding is greater for ductile materials than for fragile materials [51].

In conclusion, top milling does not appear to be favorable for surface integrity. An increase in feed rate increases roughness but reduces the internal compression stresses. Increasing cutting speed increases burrs but reduces roughness. Increasing the diamond grain size increases roughness and the density of dislocations present in the microstructure. When grinding, roughness and damage to the sublayers depend on the shaped material.

### 4.2. Surface Integrity–Functionalities Correlation

The quality and performances of parts are directly related to the surface integrity obtained by machining [52,53]. A review of the literature permits identifying the recurrent functionalities expected for a part and the components of surface integrity with which they are associated (Table 1).

**Table 1.** Components associated with functionalities expected.

| Functions | Components | References |
|---|---|---|
| Wear | Hardness, internal stress, roughness, microcracks | [10,38,54] |
| Rubbing Lubrification | Roughness | [55] |
| Sealing | Roughness | [10,38] |
| Light reflection | Roughness | [54,56] |
| Bonding | Roughness, porosity, hardness, microcracks | [10,38,54,57] |
| Fatigue | Microcracks, roughness, internal stress | [38] |
| Corrosion | Microcracks, roughness, internal stress | [38,54,57] |
| Wettability | Microstructure | [54,57] |

This review emphasizes that a set of components is necessary to characterize a functionality. Each component has a level of representativeness (relative weight) in different functionalities. Thus, different weights can be allocated to the same component when it is involved in several functionalities.

## 5. Application to Dentistry: The Surface Integrity of a Fixed Dental Prosthesis

The surface integrity of a fixed dental prosthesis obtained by CAD-CAM can be considered as the residual signature left by the machining process implemented and influencing the quality of the prosthetic surface obtained. Its surface and sublayers can be characterized according to different components linked to the clinical functionalities expected by the practitioner and the patient. The characterization of the surface of a fixed dental prosthesis relies on a number of components and multiscale and multiphysics parameters that interact with each other. Surface integrity is the intermediate concept that is indispensable between the shaping of the prosthesis and the clinical prosthetic functionalities expected (Figure 4), since a direct correlation of "process–functionalities" (correlation II) is not possible. The correlations on either side of surface integrity are highlighted by the clinical functionalities.

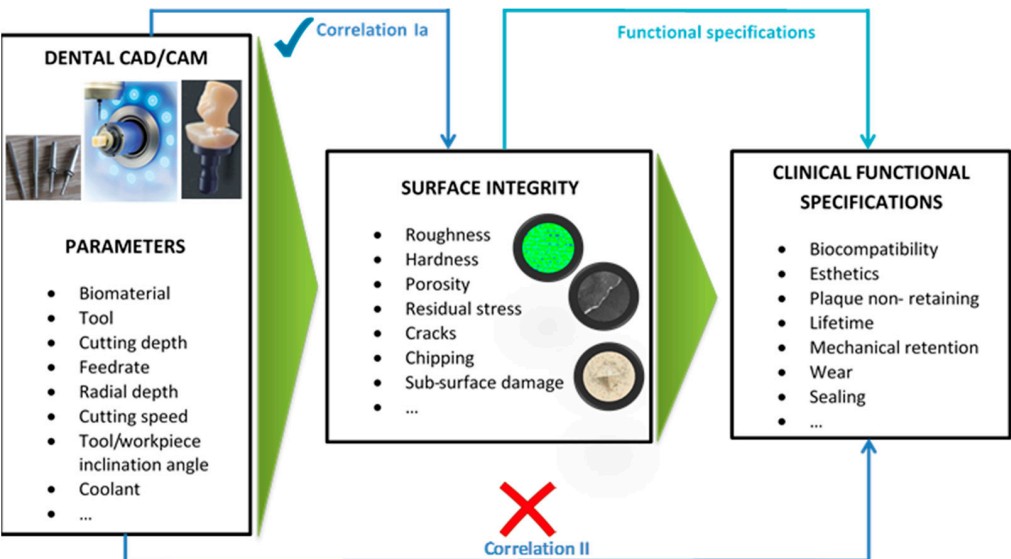

**Figure 4.** Correlations possible with surface integrity in restorative dentistry.

### 5.1. Surface Integrity–Prosthetic Functionalities Correlation

A study of the existing literature on this subject permits identifying and formalizing the components and parameters as well as several measurable indicators, representing the expected clinical prosthetic functionalities as well as possible.

### 5.1.1. Mechanical Retention

Pull-out tests permit identifying the factors that provide the best mechanical retention of a prosthesis on remaining dental tissues. Few results are available regarding non-metallic materials. However, in the case of metal crowns, the authors agree that the mechanical retention strength between the prepared tooth and the crown depends on the roughness of the surfaces sealed [58–60]. High roughness ensures better mechanical microbonding. Whatever the materials in contact and the cements used, the bond strength is proportional to the roughness parameter Rt (total roughness corresponding to the total height of the profile with the length of evaluation) [59].

### 5.1.2. Development of Dental Plaque

Studies have shown that hard dental tissues with rough surfaces favor bacterial development [61–63]. The rougher the surface is, the faster the bacterial colonization occurs, since the valleys present on the surface and linked to roughness initiate the development of dental plaque. Below a threshold of Ra = 0.2 µm, roughness no longer has an impact on the attachment of bacterial plaque [64]. Furthermore, no significant difference has been observed concerning the accumulation of plaque on surfaces with an arithmetic roughness Ra between 0.7 and 1.4 µm [63].

### 5.1.3. Optical Properties

Generally, a rough surface generates greater irrigation of the surrounding tissues, and an immediate loss of esthetic quality [35]. A surface with imperfections produces a diffuse reflection, giving a matte appearance [65]. Indeed, the surface can be likened to an infinity of tiny surfaces whose orientations vary and which therefore reflect the rays in multiple directions. Roughness permits characterizing surface imperfections. The size of the imperfections must be at least equal to the wavelength of the incident ray to be significant. The imperfections of a reflective surface measure less than 1 µm, since this is the magnitude of the wavelength of visible light [66]. The brilliance of a surface plays a major role in the esthetics of a restoration [67]. Certain studies have established a decreasing linear or logarithmic regression between brilliance values and arithmetic roughness parameters [56,68,69].

The Ra parameter has a significant influence on luminosity (the most important parameter from the optical viewpoint) and on saturation. They both increase when roughness decreases. These influences also depend on the materials.

Generally, 2D roughness influences the color of composite resins [56,70]. No clear trend has been found regarding the variation of shade linked to roughness [56].

### 5.1.4. Adherence of the Cosmetic Layer

Biaxial bending tests on bilayered disks and the theory based on adhesion energy conclude that a low roughness at the interface of two adhered or sealed disks increases the performance of the adherence between them [71–75]. Indeed, the real contact surface is directly linked to roughness Rt and it is maximal for low roughness. The smaller the real surface in contact is, the less efficient the bonding is.

### 5.1.5. Wear of a Fixed Dental Prosthesis

Fixed dental prostheses are subject to wear during friction with the antagonist tooth of the prosthesis. This phenomenon appears to be linked to the porosity of the biomaterial and the roughness of the surfaces in contact. The edges of the pores emerging at the surface and roughness peaks on the surface of the biomaterial behave like an abrasive geometry, resulting in friction generating wear of the antagonist tooth or prosthesis [76,77]. It has been shown that higher roughness leads to greater wear. A "smooth" prosthetic occlusal surface (surface in contact with the antagonist tooth when the patient closes their mouth) is therefore preferable to minimize the wear of the antagonist tooth or prosthesis [35].

5.1.6. Lifetime/Fatigue of a Fixed Dental Prosthesis

Cracking quickly leads to the destruction of dental prostheses. The prosthesis ruptures when the crack reaches a free edge. A polishing of the prosthesis that limits the risk of cracks as much as possible and generates compressive stresses that favor the closing of the cracks is therefore recommended [78].

In brief, a bibliographical study of the correlation between dental surface integrity and prosthetic functionalities emphasizes the importance of roughness in the functionalities expected. Most of the functionalities require the smoothest surfaces possible, mainly when the fatigue resistance of the prosthesis becomes an important parameter. However, it may be preferable to obtain a rough surface [51] when adhesion or retention are the functionalities expected, as with intrados of dental prostheses.

*5.2. Correlation between Fixed Dental Prosthesis CAD-CAM Process and Surface Integrity*

The residual signature of the fixed dental prosthesis manufacturing process is often approached by observing the degradation of the surface integrity components of the machined biomaterials. However, surface integrity not only depends on the finishing operation used, but also on the results of all the previous operations. The finishing operation produces only the surface texture [79]. Thus, the surface integrity of a dental prosthesis evolves and is also modified after its fabrication. From the design to the use of the dental prosthesis in the mouth, a considerable number of factors grouped into four categories are involved in modifying the initial surface integrity resulting from shaping (Figure 5). The first category acting on the surface integrity of a fixed dental prosthesis corresponds to the prosthetic material selected and its method of synthesis (the production of a raw CAD-CAM block). This mainly includes the choice of the prosthetic material and the parameters linked to its synthesis. Certain factors linked to the fabrication of a material linked to heat treatment cycles are pertinent only for ceramic prostheses (pre-sintering/sintering) [80,81].

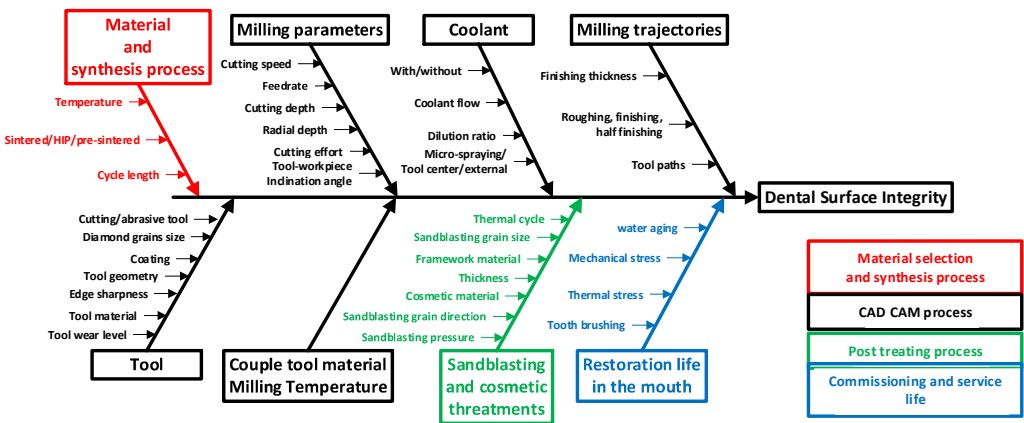

**Figure 5.** Synthesis of the main factors affecting dental prosthetic surfaces.

The second category groups factors relating to the manufacturing process embedded in a dental prosthesis CAD-CAM system. Dental CAD-CAM is the first process affecting the surface integrity of fixed prostheses. Publications on the abrasion of dental biomaterials using handpieces or turbines emphasize that the abrasive tool employed and especially the size of the diamond grains impact the formation of microcracks, chipping and roughness [82,83]. A coolant effect and cutting efforts can also cause microcracks (thermal shock) or phase changes [84,85]. Phase transformations, chipping and roughness can evolve as a function of the machined prosthetic biomaterial [82,86]. The machining parameters (cutting speed, feed rate, cutting depth, radial engagement, cutting effort) appear to play a major role in surface integrity [87–94].

The factors corresponding to actions performed after machining (sandblasting and cosmetic treatments) are grouped in the third category. This comprises the second process

affecting the surface integrity of a fixed dental prosthesis. The sandblasting of prosthetic parts essentially modifies roughness and internal stresses. The main influential factors are grain size, pressure and the orientation of the projection of grains on the surface [95,96]. Cosmetic treatment is also important. Regarding ceramic prostheses, the major factors involved are the choice of infrastructure/cosmetic pair, the thicknesses of the bilayer and the parameters of the thermal cycle used for firing the cosmetic layer. The consequences mainly occur in the internal stresses [97,98].

The last category groups the parameters affecting the surface integrity of the fixed prosthesis during its life in the mouth of the patient where surface integrity is constantly and slowly modified after machining and the secondary processes. These parameters include hydric aging, which influences internal stresses and crack propagation, the effects of tooth brushing, which modify roughness, thermal stresses linked to the temperatures of foods and mechanical stresses induced by chewing cycles that also affect internal stresses and crack propagation.

Most of the factors affecting dental surface integrity are due to the milling process embedded in CAD-CAM systems.

## 6. Conclusions

This article presented the general concept of surface integrity and its application to fixed dental prostheses shaped by CAD-CAM. These prostheses must conform to several clinical functionalities expected by the practitioner and the patient, such as mechanical retention on the tooth, the non-retention of dental plaque, the wear of the prosthesis and the antagonist tooth and the lifetime/fatigue of the prosthesis. The study of the correlation between surface integrity and the prosthetic functionalities established that roughness corresponds to a preponderant component of surface integrity in dentistry and that it has at least one correlation with all the clinical functionalities expected from a fixed prosthesis. The correlation between prosthetic surface integrity and the machining process highlighted the need to understand the mechanisms involved in dental CAD-CAM so as to improve prosthetic surface integrity and finally improve the quality of fixed dental prostheses.

**Funding:** This research received no external funding.

**Institutional Review Board Statement:** Not applicable.

**Informed Consent Statement:** Not applicable.

**Data Availability Statement:** Not applicable.

**Conflicts of Interest:** The authors declare no conflict of interest.

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
