# Peer review of "Milled Surface Integrity: Application to Fixed Dental Prosthesis"

_crystals, doi:10.3390/cryst11050559_

Round 1

Reviewer 1 Report

The paper considered an important topic of analyses of the surface integrity in the dental prosthesis. A wide range of analyses was focused on the identification of components of surface integrity and their influence on the functionalities properties of dental prosthesis. Discussed results allow bettering understanding of the correlation of milling parameters, surface integrity, and clinical functionalities of a prosthesis.

The presented review summarises the existing area of knowledge and what is most important, indicates the factor of surface integrity – roughness which correlates with functionalities properties of a fixed prosthesis and can be controlled in the process.

Nevertheless, a minor revision of the manuscript should be done before it can be accepted for publication. Please see the comments below:

  1. Figure 3 is fuzzy. Please correct it.
  2. Row 342: please change Ra =0,2 to the Ra=0.2
  3. Figure 5 - Diagram Ishikawa overlaps the figure caption.
  4. Please consider changing the type of manuscript to Review.

Author Response

You can find attached the file.

Reviewer 2 Report

This article presented the general concept of surface integrity and its application to fixed dental prostheses shaped by CAD-CAM. Its English writing is standard and accurate, and its research innovation is obvious. I recommend accepting it after making minor revision.

Author Response

You can find attached the file.
